# A Model of Angle Measurement Using an Autocollimator and Optical Polygon

Roman A. Larichev and Yuri V. Filatov *

Department of Laser Measurement and Navigation Systems, Saint Petersburg State Electrotechnical University "LETI", 197022 St. Petersburg, Russia; kerrang87@gmail.com
* Correspondence: yvfilatov@etu.ru; Tel.: +7-921-7468702

**Abstract:** The significance of an autocollimator in angular metrology cannot be overestimated: in many countries, it is either included as part of the primary plane angle standard or is involved in transferring the unit of plane angle from this standard to less accurate measuring instruments. This paper presents a historical overview of the problems encountered when using an autocollimator in angular metrology, as well as of proposed solutions. Not for the first time, the problem of the theoretical definition of the angle being measured between surfaces that are not perfectly flat is raised. In addition, the authors attempt to compile a complete list of factors affecting angular measurements using an autocollimator and to build a model that allows some of these factors to be taken into account for a subsequent algorithmic compensation of their influence. To assess the level of accuracy of angular measurements at which the use of the proposed model is reasonable, a simplified simulation example is presented. In an attempt to confirm the validity of the proposed model, a corresponding analysis of experimental data is provided. The applicability and limitations of the proposed model are discussed in the conclusion.

**Keywords:** autocollimator; angle metrology; measurement model; reflecting surface topography

## 1. Introduction

Angle measurement is an integral part of modern science and technology. It has had a vast array of applications since the dawn of science: from navigation to machinery and precise instrumentation. Currently, optical angle measurement methods may be considered among the most accurate. These methods provide measurements of a flat reflective surface's angular position. The methods are based on autocollimation or interference principles.

The autocollimation principle is represented in a wide range of industrially manufactured autocollimators [1]. This type of device is the most common one for angle measurements and varies significantly in terms of metrological performance and price. The autocollimator's applications include the measurement of linearity and parallelism of machine axes and rails, surface planarity measurement, the inspection of optical components during fabrication and assembly, etc. Depending on the task, autocollimators may provide accuracy from a few arc seconds up to a few hundredth parts of an arc second. The most accurate autocollimators may be incorporated into high precision goniometric systems, which represent primary angle standards.

Interferometric angle measurement methods are not as widespread as autocollimators for many reasons. First, they are less versatile because of high sensitivity to ambient conditions. This narrows down possible application areas to facilities with highly stabilized environment, such as measurement laboratories. The second reason is that interferometric measurement systems are more expensive compared to autocollimators. Due to the above, interferometers are suitable only for some specific angle measurement tasks. Speaking of specific interferometric angle measurement devices, the Fizeau interferometer should be pointed out [2]. This device is primarily used to measure the topography of the reflecting

surface. However, it provides an angle measurement accuracy comparable to that of a high-precision autocollimator, despite the fact that angle measurement is its secondary function.

Measurement traceability is provided through metrological routines where a less accurate measuring tool is calibrated against a more accurate standard. Thus, the unit of physical quantity is transferred from the primary standard down the chain to working measuring instruments. As for angle measurements, high-precision autocollimators are at the top of this calibration chain, either being incorporated in the primary standard, or representing one of the first links. Each national metrology laboratory has its own primary standards and performs different calibration procedures. The distribution of the plane angle quantity is performed via transfer standards. Commonly, they are represented by optical polygons and angle gauges. Optical polygons (also known as polygon mirrors) have a broad range of applications. The most recent and extensive studies on polygon mirrors were conducted in relation to the subject's application as a part of various scanner systems [3,4].

At the end of the 20th century, due to the overall increase in angle measurement accuracy, comparison measurements have been organized to verify the accuracy of angle measurement traceability in different countries. These comparison measurements [5] involved an optical polygon and different types of autocollimators with auxiliary rotary systems. The analysis of this comparison showed that the difference in the results obtained at different labs exceeded the accuracy level declared by the labs themselves. This led to the conclusion that angle measurements involving high-precision autocollimators require thorough research.

This paper gives an overview of research conducted over the past 35 years on angle measurements involving high-precision autocollimators. The authors attempted to compile all possible factors that may affect angle measurements performed by the autocollimator at an accuracy level higher than one tenth of an arc second. Furthermore, considerations are proposed regarding these factors' contribution to measurement error. A modeling case is presented to assess the influence of the reflecting surface non-flatness.

## 2. Historical Overview

The first angle measurement comparisons ran from 1980 to 1986 and involved 11 national metrology facilities from different countries. Two 12-sided optical polygons were used as transfer standards. The results that were published in the report [5] indicated that the difference between the measured values reached up to 0.3 arc seconds. The measurement accuracy stated by the comparison participants was higher than 0.1 arc second. According to the report authors, the main reason for this inconsistency was the inadequate alignment of the autocollimator-measuring axis against the polygon rotation axis.

R. Probst made a hypothesis [6,7] that the non-flatness of a reflecting surface, along with the projecting properties of the autocollimators, significantly influences angle measurements. An experiment was conducted to assess the influence of the reflecting surface non-flatness. During the experiment, the deviations from the normals' basic positions to the reflecting faces of an optical polygon were evaluated. The basic positions were defined as for the normals to the corresponding ideal regular polygon. The relative position of the real and the ideal polygons, in turn, is defined by the condition that the total of deviations over the entire polygon would be equal to zero. Angle measurements were performed both using an autocollimator and a phase-shifting interferometer. Additionally, the phase-shifting interferometer was used to measure the reflecting face topography of all polygon faces. Ultimately, the difference between the deviations obtained by devices of different types was compared to the non-flatness of the corresponding faces. The root mean square (RMS) deviation of the face points from the approximating plane was used as a quantitative measure of the non-flatness. The experiment had two test objects: a 12-sided and a 24-sided optical polygon. The results obtained for the 12-sided polygon showed that for faces with RMS less than 6 nm, the absolute difference between deviations was less

than 0.02″. Yet, the results for the 24-sided polygon were poorer: for faces with RMS less than 5 nm, the absolute difference between deviations did not exceed 0.2″. However, the author concluded that the results are in good agreement with the non-flatness, and the great difference between the results for the two polygons may be explained by the different sizes of reflecting areas. In the case of the 12-sided polygon, the reflected area was limited by a 23 mm circular aperture, and for the 24-sided polygon, the aperture was of 18 mm in diameter.

The next international angle measurement comparisons started in 1996. The PTB was the coordinating laboratory and R. Probst was the actual coordinator. During the comparisons, all participants performed calibration on a 7-sided polygon with 15 mm circular faces and a 24-sided polygon with rectangular faces 20 × 25 mm. Each lab evaluated the deviations of normals to polygon faces from their basic position applying one of two procedures described in [8,9]. The coordinating laboratory also performed measurements of polygon face topography by means of a phase-shifting interferometer. The results presented in the report [10] showed the evaluated deviations from values obtained by averaging over all participants for each face. The biggest differences (up to 0.5″) relate to those labs which either applied a certain model of the autocollimator, or placed an aperture in front of the polygon faces.

In an article [11] published in 2000, the author O. Krueger presented an experimental study on the relation between the deviation of the autocollimator measurements and the non-flatness of the reflecting faces, the polygon eccentricity and the pyramidal error. The experiment consisted in calibrating the autocollimator against the interferometric small-angle generator within a 40 arc second range, using five mirrors of different topographies. The repeatability of the small-angle generator positioning was better than 0.02″. Generally, all mirrors were evenly concave or convex, and their non-flatness was quantified by the RMS and peak-to-valley parameters of the surface. Compared to the flattest one, the deviations of the calibration values obtained with different mirrors reached 0.09″. Moreover, deviations for concave and convex mirrors had opposite signs. Upon analyzing the results, the author suggested that the measured value's deviation from the actual mirror turn angle caused by the non-flatness of the mirror may be introduced by the formula $u = k \cdot z \cdot x$, where $u$ is the deviation, $x$ is the mirror turn angle, relative to the autocollimator axis, $z$ is the RMS of the reflecting surface topography and $k$ is the scale factor specific for the current autocollimator. Therefore, the angle measurement deviation was in direct proportion to the RMS parameter of the mirror. However, it did not explain the difference between the signs of the deviations for convex and concave mirrors. In fact, the RMS parameter does not account for the actual shape of the surface and may be identical for concave, convex or any other mirrors of more complicated topographies.

The eccentricity influence was investigated by measuring the mirror's angle deviation at its different lateral shifts in the ±0.5 mm range. All shifts had to be parallel, and therefore they were controlled by the same small-angle generator. The measured deviations did not seem to correlate with the mirror's topography. At a 0.1 mm shift, the deviation reached 0.04″, and at 0.5 mm it reached 0.12″. The author deemed all the deviation values presented in the article as negligible. The next international angle measurement comparison started the same year. The coordinating laboratory was from South Africa, with O. Krueger in charge [12]. A 12-sided optical polygon and a set of four wedges with different nominal angles were used as transfer standards. There were no significant scientific results or theories suggested in the report. The results provided by the participants were in close agreement considering the polygon measurement, yet the wedge measurements did not match so well. The reason for this, as stated by the authors, was that few laboratories provided the same measurement uncertainty for the wedge measurement as for the polygon, although it had to be greater. These are the first comparisons the report of which included sections providing insight on the evaluation of measurement uncertainty. The definition of the optical polygon pitch angles that were measured and subsequently compared is presented in Figure 1.

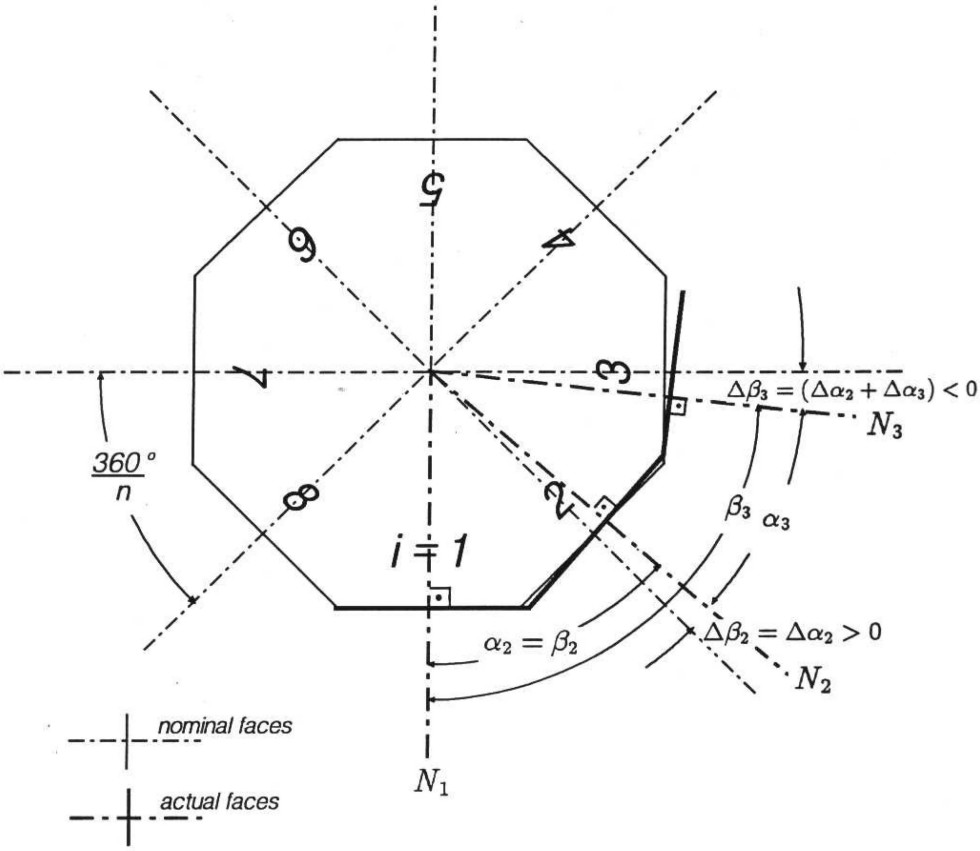

**Figure 1.** Schematic drawing of the polygon.

The pitch angles $\alpha_i$ are the angles between the projections of two adjacent normals $N_{i-1}$ and $N_i$ in the measuring plane with the counting index ($i = 1, 2,\ldots, n$). The deviations of the pitch angles from their nominal values of $360°/n$ are referred to as pitch angle deviations ($\Delta\alpha_i$). $\beta_i$ and $\Delta\beta_i$ are cumulative polygon angles and their deviations, respectively.

Prior to the comparisons via technical protocol, all participants were provided with a sample comparison report relating to the long gauge linear measurements performed by interferometric methods [13]. The participants were also free to develop and apply their own uncertainty evaluation methods that had to comply with the ISO Guide for the Expression of Uncertainty in Measurement. Such a freedom was justified by the diversity of the calibration setups.

However, the majority of participants utilized the same measurement model to submit their uncertainty budgets:

$$\Delta\alpha_i = \alpha_i - \frac{360°}{n} + \delta A_F + \delta A_P + \delta A_E \qquad (i = 2, 3 \ldots n)$$
$$\alpha_i = \text{Ç} - \S \tag{1}$$

where $\delta A_F$—correction for flatness deviations of the measuring face, $\delta A_P$—correction for pyramidal errors of the measuring face, $\delta A_E$—correction for eccentricity errors in the setup of the polygon/angle block, §—autocollimator/interferometer reading, Ç—index table reading and $i$—measuring face index.

The uncertainty contributions corresponding to expression members Ç, §, $\delta A_F$, $\delta A_P$ and $\delta A_E$ were listed in the report as declared by the participants. All contributions were given as standard uncertainties together with the calculated combined uncertainty and repeatability in a summary table (Table 1).

**Table 1.** Uncertainty contributions to measurements of optical polygon pitch angles given in milliarc-seconds.

| Laboratory | § | Ç | $\delta A_F$ | $\delta A_P$ | $\delta A_E$ | Repeat. | Combined Uncertainty |
|---|---|---|---|---|---|---|---|
| NIM (China) | 29 | 17 | 47 | 0.4 | 47 | 2.4 | 73 |
| KRISS (Korea) | 30 | 40 | 15 | 36 | 9 | 25 | 69 |
| NMIJ (Japan) | 12 | 8.6 | | | | 10 | 18 |
| SMU (Slovakia) | 10 | 60 | 20 | | 1 | | 60 |
| PTB (Germany) | 10 | 2.4 | 4 | 9 | | 3 | 15 |
| METAS (Switzerland) | 10 | 20 | 80 | 30 | | 20 | 90 |
| LNE (France) | 10 | 15 | 4.7 | 4.7 | 4.7 | 28 + 4.7 Auto + Table | 30 |
| IMGC (Italy) | 10 | 6 | 10 | | | 5 | 16 |
| NIST (USA) | 13 | | 9 | 4 | 4 | 10 | 20 |
| CENAM (Mexico) | 35 | | 31 | 80 | 4 | 7 | 82 |
| NRC (Canada) | 30 | 20 | 50 | 25 | 25 | 25 | 75 |
| VNIIM (Russia) | 20 | 30 | 20 | 30 | 10 | 30 | 60 |
| CSIR-NML (South Africa) | 25 | 50 | 28 | 25 | 10 | 30 | 75 |

According to this table, a few participants have omitted certain contributions or deemed them as negligible. As there is no information on the estimates of $\delta A_F$, $\delta A_P$ and $\delta A_E$ corrections themselves, it may be concluded that those factors were taken into account only as sources of random error. In addition, there are no comments on the uncertainty evaluation methods applied by the participants. Therefore, it appears that all participants agreed to the model (1), yet they evaluated the uncertainty contributions using their own methods. Although the deviations in uncertainty evaluation methods were sanctioned due to the diversity of calibration setups, they raise doubts regarding the consistency of the results submitted by different laboratories. The sample report provided to the participants prior to the comparisons references article [14], which presents a very elaborate model for gauge block length measurements based on interference. This model describes the influence of multiple factors on the measurement process, and thus defines the way each factor should be accounted for in evaluating measurement uncertainty. On the other hand, the origin of the applied model used for angle measurements is unknown and the model itself is limited by expression (1), lacking any details. This issue remained unsolved over a considerable period. Two comparison measurements [15,16] took place, until the comparison measurements conducted in the SIM region in 2008–2009 [17]. The report on the latter eventually pointed out the issue of the angle measurement model and the need to develop a precise and unambiguous one. However, such a model has not yet been developed.

## 3. Materials and Methods

### 3.1. Factors Influencing Autocollimator Measurements

Before attempting to address the measurement model directly, it would be prudent to specify the factors affecting autocollimator measurements that are already known. International metrology is not the only area of research that involves high-precision autocollimator angle measurements. While researching deflectometric measurements of a synchrotron optics with an accurate digital autocollimator, the German authors have already selected a significant amount of factors of interest [18]. All these factors were divided into two groups: external and internal.

External factors:

- Reflectivity of the surface under test (SUT);
- Curvature of the SUT;
- Distance (path length) to the SUT;
- Diameter and shape of the aperture stop;
- Position of the aperture stop along the autocollimator's optical axis;
- Lateral position of the aperture stop perpendicular to the optical axis.

Internal factors:

- Aberrations of the optical components (autocollimator's objective, reticle illumination, beam splitter cubes, etc.);
- Alignment of the optical components and of the CCD detector;
- Non-orthogonality of the measuring axes;
- Reflections from internal optical components;
- Geometrical imperfections of the reticles;
- Inter-pixel variations of the CCD (geometry, quantum efficiency, dark current, etc.);
- Intra-pixel quantum efficiency pattern (across single CCD pixels, due to their internal structure).

The internal factors present the most difficulty, as they are impossible to assess. High-precision autocollimators are industrially manufactured devices. This implies that they cannot be taken apart to be studied, unless the manufacturer participates in the research. The list of external factors should also include ambient air pressure fluctuations. The influence of this factor was registered and studied during the most recent international angle measurement comparisons that had the autocollimator as the transfer standard [19]. The autocollimator's focal length and therefore its scale factor are defined by the geometrical parameters of the objective lens and by the relation between the refractive indexes of lens and air. As the refractive index of air is dependent on the ambient pressure, the pressure fluctuations result in autocollimator errors. In addition, a systematic error arises due to the difference between the conditions at which the autocollimator was assembled and calibrated, and its operational conditions. Some of the presented external factors were studied experimentally, such as the distance from the autocollimator to the SUT, and the size and position of the aperture stop in the light propagation path [20]. The experiment consisted of repeatedly calibrating the autocollimator against the high-precision index rotary table. One of the parameters was modified under research. The results have shown that these parameters can cause deviations in autocollimator readout of approximately 0.02–0.03 arc seconds. Additionally, it was mentioned that regarding the position of the aperture stop along the autocollimator optical axis, the optimum performance was achieved when the aperture stop was the closest to the SUT. j

### 3.2. Angle Definition

There is also a fundamental problem related to angle measurements that should be addressed before any model is developed. It is the problem of angle definition, taking into account the curvature of the SUT. In goniometry or angle measurements involving the autocollimator, the quantity that is measured is typically an angle either between two reflecting surfaces, or between the normal to the reflecting surface and the autocollimator optical axis. In both cases, it is assumed that the surfaces are ideal planes, and the measurand corresponds to its mathematical definition. In practice, every reflecting surface has a slight curvature, along with fabrication flaws, which makes it impossible to use it as direct reference for angle measurement. From this point forward, the problem splits into two: the first one refers to a quantity that is actually measured by the autocollimator, and the second one consists in defining what should be measured taking into account the surface curvature. The key to understanding both these problems is replacing the real SUT by the approximating effective plane. It is clear that the autocollimator does not physically replace the SUT; instead, it registers the surface as if it were a plane. The way the surface is approximated by the effective plane in this case is determined through the autocollimator operation principle. There are multiple mathematical methods of approximating a surface with a plane. Some of them have already been studied in relation to the angle measurement problem. In 2004, a group of American authors published an article [21] in which they considered various surface approximation methods (see Figure 2) and grouped them by the instruments that utilize them.

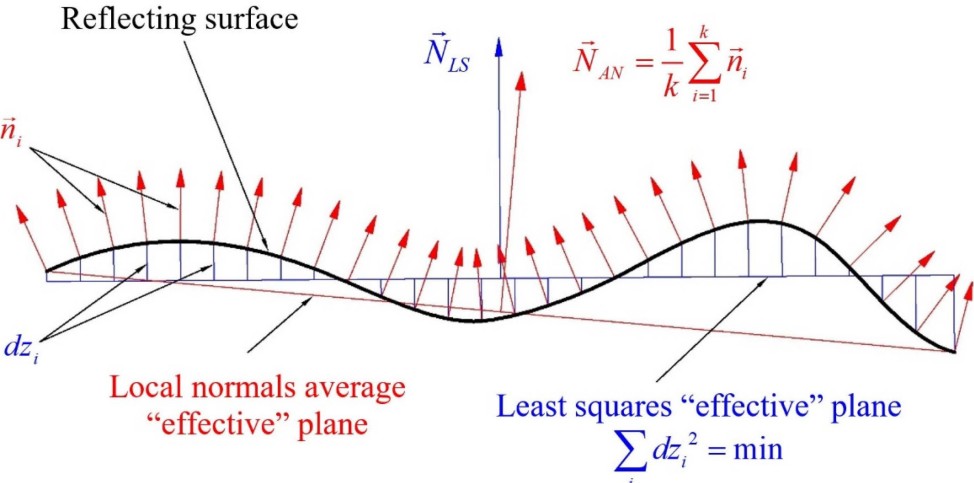

**Figure 2.** The approximation methods for non-flat surfaces.

The first one was the least squares method which is applied in the Fizeau phase-shifting interferometer (PSI). In Figure 2, $\vec{N}_{LS}$ is a normal which refers to the effective least squares plane. The PSI is not an angle measuring instrument, but its main purpose is to measure the curvature of the reflecting surface. However, when the surface is measured pointwise, the PSI software expands the curvature in terms of Zernike aberration polynomials of the first 36 orders. The second and the third members in this expansion characterize the general inclination of the effective least squares plane to the two orthogonal axes that are associated with the measured surface.

The other approximation method consisted in averaging the local normals to the surface. Thus, the effective plane was found as the orthogonal to the average normal $\vec{N}_{AN}$. The authors analyzed the autocollimator's optical scheme from the perspective of ray geometry, and have shown that this approximation method is in agreement with the operation principle of the autocollimator. The assumptions of the autocollimator model used by the authors were as follows. The illuminated reticle was the point light source, the illumination of the autocollimator exit aperture was uniform, and the position of the

reticle image in the focal plane was found as the illumination center of mass. Another important conclusion following this analysis is that the angular position of the effective plane does not depend on the shape of surface curvature, but only on its border. This is implied by the approximation method itself: the angle of the local normal is found as the partial derivative of the surface function. In order to obtain the angle of the average normal, all local normals are integrated across the surface. However, if the surface is continuous, the integration product will solely depend on the surface function values at the integration limits. And these limits are at the surface border. The analysis of the average normal approximation method not only answers the question of what the autocollimator measures, but also reveals the simple autocollimator measurement model that takes surface curvature into account. Thus, the correct autocollimator measurement model, the necessity of which was pointed out earlier, could be developed on the basis of local normals' approximation method by adding the parameters listed in the previous section.

### 3.3. The Autocollimator Angle Measurement Model

The inability to assess and study the internal factors that affect the autocollimator measurements makes it impossible to develop a comprehensive measurement model. Thus, the only way left is trying to approach it as closely as possible by introducing certain assumptions and by considering the external factors.

An autocollimator with the reflecting surface is an axisymmetric optical system, which projects the image of the illuminated reticle to the autocollimator focal plane. The angle $\varphi$ between the autocollimator's optical axis and the normal to the reflecting surface is found as:

$$\varphi = \frac{\arctan(x/f)}{2} \tag{2}$$

where $x$ is the lateral shift of the reticle image relatively to the intersection point of the autocollimator axis and the focal plane, and $f$ is the focal length of the lens. What follows from this expression is that the angle measurement error may be the product of the focal length deviation. At the same time, it can come from the error made when determining reticle image displacement. It has already been pointed out that the focal length fluctuation of the objective lens is a function of the air refractive index, presuming that the geometry and refractive index of the lens are constant. And the refractive index of air mainly depends on air pressure. In any case, errors associated with the fluctuation of the focal length have been discussed substantially in paper [22] and are not the point of interest of the current work. On the contrary, the reticle image displacement is the parameter that influences all factors under research.

The optical scheme of the setup that comprises an autocollimator and reflecting surface will be further considered from the perspective of wave optics. The main assumptions regarding the optical scheme will be the same as in the model earlier: the reticle is a point light source, the illumination is uniform across the exit aperture and the position of the reticle image is the illumination center of mass. The impact of the autocollimator's internal parameters will be introduced as an arbitrary distribution of wave aberrations concentrated at the exit aperture of the autocollimator. The impact of the reflecting surface non-flatness may also be considered as the aberration that it brings to the autocollimator's wavefront. It is convenient to operate with wave aberrations in terms of Zernike polynomials. These polynomials are orthonormal and defined on the unit circle. Thus, the whole distorted wavefront may be expanded as a linear combination of aberrations defined by separate Zernike polynomials (see Figure 3).

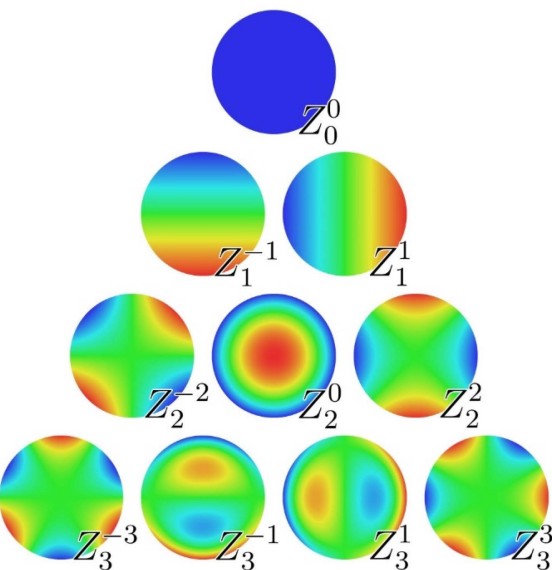

**Figure 3.** Zernike polynomials up to the third order.

The main goal of the model applying the wave optics approach is to assess how certain Zernike aberrations affect the estimated position of the point light source image. The fact that Zernike polynomials are orthonormal allows researchers to investigate the influence of different aberrations separately. The axisymmetric optical system projects the point light source into an intensity distribution called the Airy pattern. The image aberration theory in respect of such systems [23] has indicated that most types of aberration distort the image in a symmetrical way without shifting the intensity center of mass. The exceptions are the coma aberrations of all orders and the aberrations that represent the general wavefront inclination (see Figure 4).

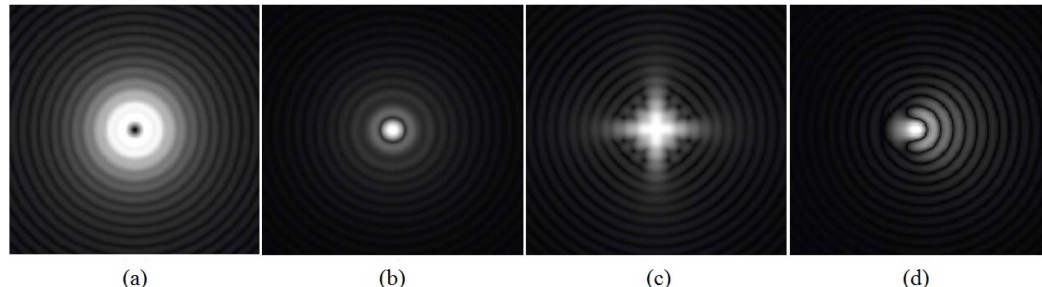

|     |     |     |     |
| --- | --- | --- | --- |
| (a) | (b) | (c) | (d) |

**Figure 4.** PSFs (point spread functions) of the optical system referring to certain Zernike aberrations: (**a**) defocus, (**b**) spherical, (**c**) astigmatism and (**d**) coma.

This is in agreement with the method for averaging local normals introduced earlier. It was shown that the autocollimator which operates according to that approximation method, in fact, measures the inclination of the reflecting surface border. Along with this, it appears that only the general inclination (the 2nd and 3rd members in the Zernike expansion) and coma aberrations have their border tilted relatively to the base reference plane. Considering coma and inclination aberrations, the theory analysis has shown that the shift occurring in the intensity distribution centroid of the point image is proportional to the value of the Zernike aberration. The value of the aberration is defined by the corresponding coefficient. Taking into account Equation (2) and a small value of measured angles, it may be concluded that the aberration value is linearly proportional to its angle response.

## 4. Results

### 4.1. Model Development

Up to this point, the analysis of the aberrations was conducted without considering the propagation of the wavefront. Further, the impact of the external and internal factors influencing the autocollimator measurements on the shaping of the aggregate wavefront will be considered. Figure 5 represents the unfolded optical scheme for the basic case where the reflecting surface is orthogonal to the autocollimator axis, and both the autocollimator exit aperture and the reflective surface are circular and of the same diameter.

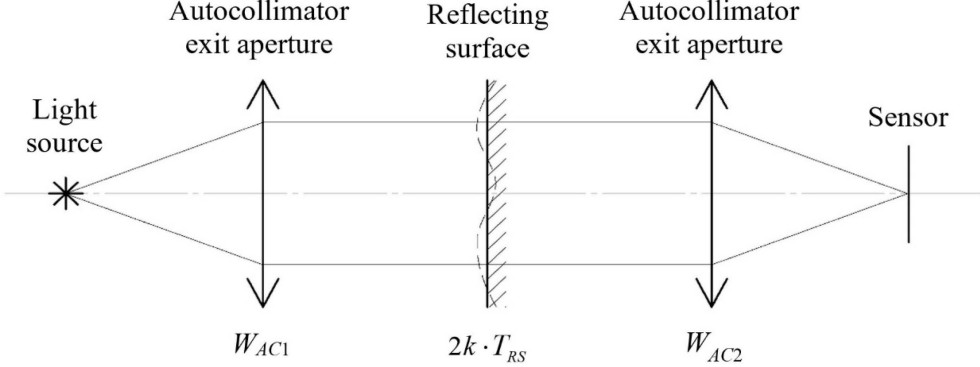

**Figure 5.** Autocollimator unfolded optical scheme.

$T_{RS}$ is the topography of the reflecting surface which is given in terms of height deviation relatively to the least squares approximating plane. The surface's contribution to the wavefront aberration caused by the reflection will be defined as $2k \cdot T_{RS}$, where k is the wave number of the propagating light. The influence of the autocollimator's internal factors is presented by two wavefront aberration distributions $W_{AC1}$ and $W_{AC2}$ given in terms of wavefront phase deviation. The ideal wavefront is transformed as $W_{AC1}$ distribution while travelling from the light source to the autocollimator exit aperture, with $W_{AC2}$ aberrations added to the wavefront on the light's way backwards from the exit aperture to the sensor. The common factor for these two distributions is the aberration brought by the autocollimator objective lens. Besides this, $W_{AC1}$ accounts for the illumination-related factors, while $W_{AC2}$ does the same for the influence of the sensing element's parameters. The contribution of the beam-splitting element toward wavefront shaping also has to be accounted for separately, as it transmits the light in a forward direction and reflects it in a reverse direction. At this point, the upcoming logic demands an additional assumption. This assumption is that the plain or nearly plain wavefront stays invariant while travelling through free space. Research [24] has shown that, in fact, the free space transformation of such a wavefront is very small, so it will be deemed negligible at this moment. Therefore, having incorporated the above assumption, the final wavefront $W_\Sigma$ may be found as follows:

$$W_\Sigma = W_{AC1} + 2k \cdot T_{RS} + W_{AC2} \tag{3}$$

The basic case and formula (3) are presented in order to demonstrate how the aggregate wavefront builds up, regardless of the external measurement factors. The common measurement case including the reflecting surface inclination and the aperture stop is presented further in Figure 6.

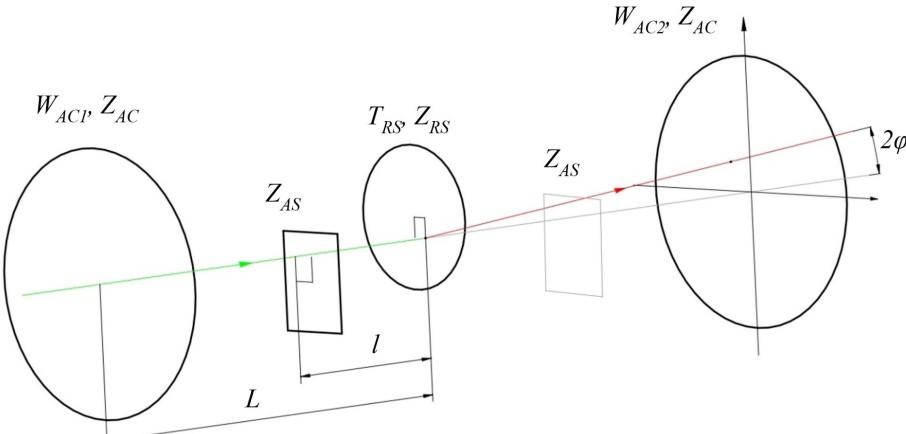

**Figure 6.** The common measurement case.

Here, the colored lines with arrows indicate the propagating light beam. The green line is for the beam before reflection, and the red is for the beam after reflection. The autocollimator is the reference unit. The 0xy coordinate system is bound to the center of the autocollimator exit aperture, at the point where the wavefront returns to it. The system axes are parallel to the autocollimator axes, along which the image displacement is measured. As the overall inclination $\varphi$ of the reflecting surface is small, it may be considered as a combination of two inclinations $\varphi_x$ and $\varphi_y$ under the condition that $\varphi^2 = \varphi_x^2 + \varphi_y^2$, where $\varphi_x$ is the inclination around 0y, and vice versa. $L$ is the distance from the autocollimator exit aperture to the reflecting surface, and $l$ is the distance from the reflecting surface to the aperture stop. $W_{AC1}$, $W_{AC2}$ and $T_{RS}$ are defined relatively to the centers of the corresponding elements. The dimensions and shapes of the autocollimator exit aperture, reflecting surface and aperture stop are introduced through the special functions $Z_{AC}$, $Z_{RS}$ and $Z_{AS}$, which are also defined relatively to their element's centers. The $Z$ function is equal to 1 for all the points that are inside the bounds of the element and equal to 0 for the rest of the points. While the autocollimator exit aperture is circular, the shape of the reflecting surface and aperture stop may be random (it is also usually circular or rectangular). The lateral displacement of the reflecting surface and the aperture stop related to the autocollimator's optical axis is set in terms of their center coordinates ($x_{RS}$; $y_{RS}$) and ($x_{AS}$; $y_{AS}$).

The inclination of the reflecting surface $\varphi$ will also cause the general inclination of the propagating wavefront which has to bring the additional member to Equation (3). Inclination $\varphi$ is, in fact, the quantity which is to be measured by the autocollimator and thus the additional member would represent the ideal wavefront. As the measurement model was developed to assess the errors that cause certain factors, this member will be omitted. The aggregate wavefront $W_\Sigma$ will be substituted by the wavefront that contains only aberrations that cause measurement errors $W_{\Sigma er}$. As a result, the wavefront $W_{\Sigma er}$ will be determined by:

$$
\begin{aligned}
W_{\Sigma er}(x;y) = &[W_{AC1}(x - L\cdot\tan(2\varphi_x); y - L\cdot\tan(2\varphi_y)) + \\
&+ 2k\cdot T_{RS}(x - x_{RS} - L\cdot\tan(2\varphi_x); y - y_{RS} - L\cdot\tan(2\varphi_y)) + \\
&+ W_{AC2}(x;y)]\cdot Z_{AC}(x;y)\cdot Z_{AC}(x - L\cdot\tan(2\varphi_x); y - L\cdot\tan(2\varphi_y))\cdot \\
&\cdot Z_{RS}(x - x_{RS} - L\cdot\tan(2\varphi_x); y - y_{RS} - L\cdot\tan(2\varphi_y))\cdot \\
&\cdot Z_{AS}(x - x_{AS} - L\cdot\tan(2\varphi_x); y - y_{AS} - L\cdot\tan(2\varphi))\cdot \\
&\cdot Z_{AS}(x - x_{AS} - (L - l)\cdot\tan(2\varphi_x); y - y_{AS} - (L - l)\cdot\tan(2\varphi_y))
\end{aligned}
\tag{4}
$$

The interpretation of this expression may be as follows. Three different distributions of the wavefront aberration are summed up depending on the overlap caused by their alignment properties, on the distance to the surface and on the actual angle that is being measured. Afterwards, the sum is cropped (multiplied by the $Z$ functions) depending on

the same factors. It depends also on the shapes and dimensions of the considered elements, and the axial and lateral positions of the aperture stop. Ultimately, the resultant wavefront $W_{\Sigma er}$ has to be analyzed in order to determine the shift in the projected image centroid and the measurement error, respectively. It may be distributed in terms of Zernike aberrations to find the values of inclination and coma coefficients. Alternatively, the angular measurement error may be evaluated numerically as the general inclination in the wavefront border. Although the Zernike polynomials are quite common in aberration processing, it may be unfavorable to apply them to the problem at hand. This is mainly due to the area of the processed wavefront not being constant, which is caused by the cropping defined by the variable parameters of measurement. Supposing that there is an aberration distribution of a certain area with known Zernike expansion that is limited by some arbitrary aperture, the Zernike expansion for the derivative aberration distribution can never be obtained directly from the initial set of coefficients. Therefore, to obtain the Zernike expansion for the derivative distribution, the primary distribution would have to be reconstructed from the known expansion, cropped and expanded afterwards once again. This implies making a large number of calculations. The other problem is that the actual wavefront border may not be circular. In fact, taking into account all the overlapping and cropping from expression (4), the final wavefront outline will most definitely not be circular. There is a method for non-circular aberration distribution in terms of orthonormal polynomials [25], but it also requires massive calculations.

The suggested model (4) provides an opportunity to evaluate the angular deviation of the real reflecting surface relatively to its least squares approximation plane measured by the autocollimator. This possibility is conditional upon the fact that the topography of the surface and the inner autocollimator aberrations are known, and the measurement setup parameters are defined and well controlled. Conventionally, the technical specifications for the autocollimators, provided by the manufacturers, determine both the angle measurement accuracy level with measurement range and the maximum distance to the reflecting surface. Some of these also specify the required minimum dimensions of the reflecting surface and the reflectivity. Manufacturers rarely specify the required surface flatness as the root mean square of the surface profile height. Usually, this is sufficient for common angle measurement applications that do not demand extreme accuracies. Yet, there are specific tasks such as the comparison measurements considered in the overview. Such measurements have to make use of the suggested model with all the listed parameters that influence the autocollimator operation. It is clear that these have to be the tasks that imply the major part of the influencing parameters to be constant in order to reduce the amount of numerical calculations.

*4.2. Numerical Simulation of Angle Measurements*

In order to estimate the accuracy level, at which the model application is essential, the following simplified simulation problem was considered. In principle, the deviation in autocollimator angle measurement arises due to various cropping of the cumulative wavefront. Thus, the point of the simulation was to generate a set of test wavefronts that would be close to what is found in practice. Then, the autocollimator's angle deviations should be calculated when these wavefronts were cropped. As it was mentioned earlier, the impacts of the aberrations represented by different Zernike polynomials can be investigated separately. Therefore, the generated set consisted of wavefronts represented by the first-order Zernike aberrations of defocus, astigmatism, coma and spherical aberration. Commonly, the aberrations of these types bring the major contribution to overall wavefront topography. All generated wavefronts were defined inside the circular area of 20 mm diameter. To bring the values of the Zernike coefficients into compliance with the physical dimension of the wavefronts, they were selected as $C_3 = 10$ nm for defocus, $C_4 = 10$ nm for astigmatism, $C_6 = 10$ nm for coma and $C_8 = 5$ nm for spherical aberration. The choice of these values was based on face topographies measured for the 24-sided optical polygon which is regularly used in metrological routines. It was assumed that the autocollimator's

internal aberrations do not exceed those that are caused by the non-flatness of the reflecting surface.

In order to estimate the angular deviations, each generated wavefront was masked with the circular aperture of 15 mm diameter. All deviations were calculated, applying the average normal approximation method for the cropped wavefront area to bring the result in accordance with what the autocollimator would register. The variation of the cropping area was realized by shifting the aperture relatively to the initial wavefront center along the axis of conjectural measurements (see Figure 7). The maximum aperture shift considered was ± 2.5 mm.

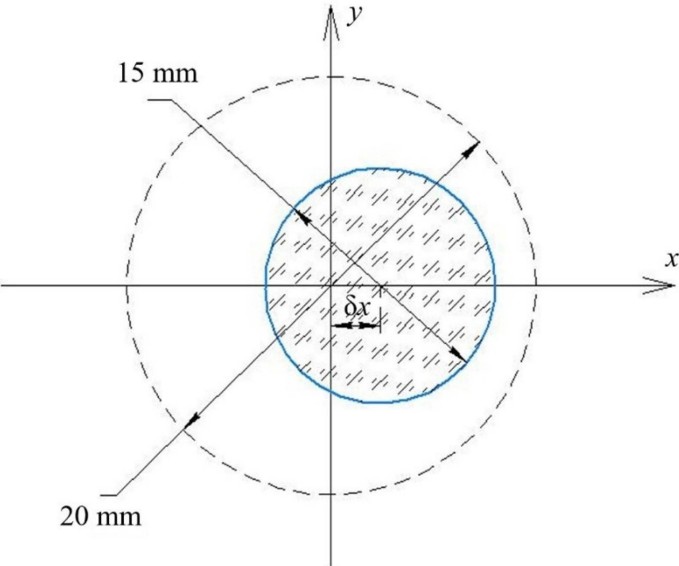

**Figure 7.** Variation of the cropping area.

All angle deviation values were calculated relatively to the initial wavefront of the least squares approximating plane. Figure 8 represents the results combined in a single chart.

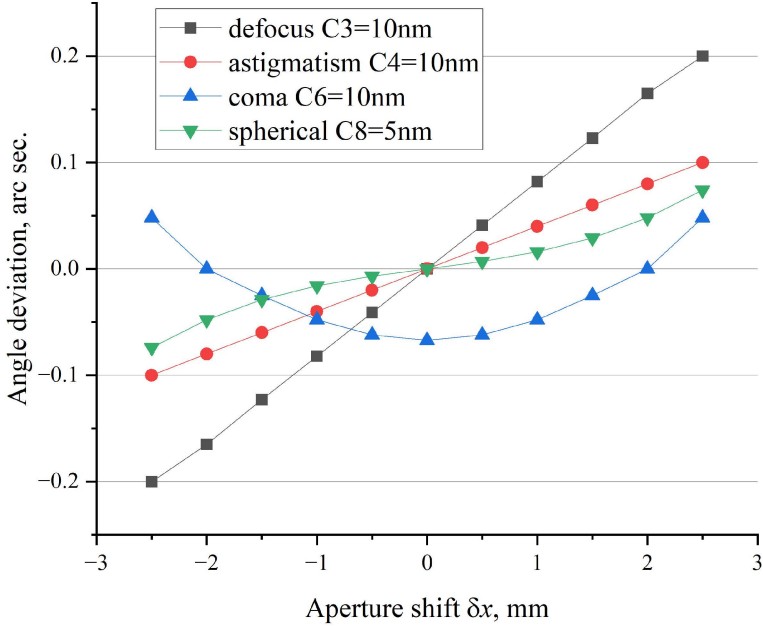

**Figure 8.** Simulation results.

Naturally, any real wavefront is a combination of aberrations considered in the simulation case and much more. As the aberrations are summed up, they may build up the cumulative angle deviation or, alternatively, they may compensate each other, depending on the sign of the corresponding Zernike coefficients. The values of each Zernike coefficient applied in the simulation problem were selected at the upper margin of used measurement results. This means that summing up the maximum values of calculated angle deviations would correspond to some extreme case, which could hardly be found in practice. Having summarized all of the above and relying on the simulation results and common sense, the authors suggest to set the sought-for accuracy level at 0.2 arc seconds.

*4.3. Experimental Verification of the Model Consistency*

A large amount of variable parameters and unknown internal aberration distributions of the autocollimator $W_{AC1}$ and $W_{AC2}$ in model (4) significantly complicate the experimental research of this model. The only method that could resolve this issue is to eliminate the unknown distributions and to make the majority of the model variables constant. In practice, both these goals are attainable with a goniometric setup for optical polygon calibration which was generally utilized during the comparison measurements. An optical polygon is a regular prism with all the reflecting faces normal to its base. Polygon calibration consists in determining the angle deviation of each face normal relatively to its position, as defined by the corresponding ideal geometrical figure. The polygon angles themselves may be expressed with reference to the first face, so-called cumulative angles, or as the pitch angles that are measured sequentially between adjacent faces. The evaluation of the corresponding deviations for these polygon angles then continues as follows:

$$
\begin{aligned}
&\Delta\beta_i = \beta_i - \frac{2\pi}{N} \cdot (i-1), \ \beta_i = \gamma_i - \gamma_1 \\
&\Delta\alpha_i = \alpha_i - \frac{2\pi}{N}, \ \alpha_i = \gamma_i - \gamma_{i-1} \\
&i = 1 \ldots N
\end{aligned}
\tag{5}
$$

where $\beta_i$ and $\alpha_i$ are cumulative and pitch polygon angles, $\Delta\beta_i$ and $\Delta\alpha_i$ are their deviations, $N$ is the number of polygon faces and $\gamma_i$ is the angle position of each polygon face measured directly by the goniometric system relatively to the coordinate system set by the angle encoder of the index table. As it can be observed from expressions (5), each angle deviation depends on the measured angle position of two faces: the current and the first ones for the cumulative angle, or the current and the previous ones for the pitch angle. In the report to one of the first international comparison measurements (the Probst comparison report), it was proposed to apply reduced angle deviations to be able to investigate the deviation referring to a single face. These deviations $\overline{\Delta\beta_i}$ are derived from the cumulative ones:

$$
\overline{\Delta\beta_i} = \Delta\beta_i - \frac{1}{N}\sum_{i=1}^{N}\Delta\beta_i
\tag{6}
$$

The reduced deviation does not depend on any specific face, besides a current one to which it refers. In order to clarify the connection between the goniometric measurements and the model verification, the angle position of each polygon face has to be taken apart by its components. The goniometric system basically consists of the index table and the autocollimator mounted on the common base. The autocollimator implements an optical connection to the polygon reflecting faces so that their angular position could be measured relatively to the angular scale of the index table. During the measurement, each face is sequentially positioned against the autocollimator, so that the angle between the face normal and the autocollimator optical axis is within the autocollimator measurement range. After the index table stabilizes while positioning one of the faces, the readings from the autocollimator and the angular scale of the index table are taken, and the angle position of the reflecting face is evaluated as their difference. Following this, the index table rotates to

position the next face. Taking into consideration the systematic errors of the autocollimator and the angular scale, the angle position of the *i*-th face may be defined as:

$$\gamma_i = (e_i + \Delta e_i) - \left( a_i^{LS} + \Delta a_i^{W_{AC}} + \Delta a_i^{W_{RS}} \right) \tag{7}$$

where $e_i$ is the current angle position of the index table presuming that its angular scale is ideally regular, $\Delta e_i$ is the systematic error of the scale for this position, $a_i^{LS}$ is the angle between the autocollimator optical axis and the normal to the least squares approximating plane of the current reflecting face, $\Delta a_i^{W_{AC}}$ is the angle deviation of the face caused by the internal aberrations of the autocollimator and $\Delta a_i^{W_{RS}}$ is the angle deviation of the face caused by its non-flatness. As a rule, in order to reduce the autocollimator measurement error, during each positioning, the polygon face is set as close to the normal position against the autocollimator as possible. Assuming that it is a sharp normal, the member $a_i^{LS}$ in expression (7) may be set to zero. Moreover, returning to expression (4), $\varphi_x$ and $\varphi_y$ may also be set to zero. For the former one, this is due to the same reason as for $a_i^{LS}$, i.e., the angle that lies in the goniometric measurement plane. For the latter one, this is due to the polygon adjustment relatively to the rotation axis and the assumption that the polygon pyramidal error may be deemed as negligible. As for the shaping of the aggregate autocollimator wavefront, the pyramidal error gives rise to a variable shift of wavefront cropping defined by the parameter $L \cdot \tan(\varphi_y)$. However, taking into account that for high-precision goniometer measurements, the distance between the reflecting face and the autocollimator exit aperture is typically below 20–30 cm and the polygon pyramidal error itself does not exceed a few arc seconds, this shift stays below 0.02 mm. Such values may indeed be considered as negligible.

The elimination of angle variables from expression (4) leads to the autocollimator's internal aberrations ($W_{AC1}$ and $W_{AC2}$). In this situation the impact of these aberrations on the measured face angle position $a_i^{W_{AC}}$ is identical for all polygon faces, except for a certain case when disposing of angle variables is insufficient. This case occurs under the following conditions: the polygon is mounted eccentrically relatively to the index table rotation axis, the dimensions of the reflecting faces are smaller than the autocollimator exit aperture, and the aperture is absent. The above conditions make the overall wavefront cropping from expression (4) totally defined by the factor $Z_{RS}\left( x - x_{RS} - L \cdot \tan(2\varphi_x); y - y_{RS} - L \cdot \tan(2\varphi_y) \right)$. The eccentricity error makes this factor variable, depending on the magnitude of the eccentricity and reflecting face sequence number via the variance of the $x_{RS}$ parameter. In practice, the eccentricity error can be reduced sufficiently by using special polygon housings and centering appliances. Proceeding to the reduced angle deviations of the polygon faces, they become independent of the autocollimator internal aberrations. Taking in account expressions (5) and (7) and all the considerations presented above, the reduced deviations may be expressed as follows:

$$\overline{\Delta\beta_i} = \left[ e_i - \frac{1}{N} \cdot \sum_{i=1}^{N} e_i \right] + \left[ \Delta e_i - \frac{1}{N} \cdot \sum_{i=1}^{N} \Delta e_i \right] - \left[ \frac{2\pi}{N} \cdot (i-1) - \sum_{i=1}^{N} \frac{2\pi}{N^2} \cdot (i-1) \right]$$
$$- \left[ \Delta a_i^{W_{RS}} - \frac{1}{N} \cdot \sum_{i=1}^{N} \Delta a_i^{W_{RS}} \right] \tag{8}$$

Here, the first two brackets relate to the angular scale readings of the index table. The third one is defined by the number of the currently measured face and the total number of polygon faces, and the last one represents the influence of the polygon faces' non-flatness on the autocollimator measurements. In order to use the measurement results for model verification, the autocollimator-related components have to be isolated. This can be achieved by running two similar calibrations of the same polygon with different apertures placed between the autocollimator and the polygon in the process. The fact that the calibrations are similar implies ruling out polygon reinstallation or changing any other measurement setup parameters, except inserting the apertures. Therefore, if

the experiment is carried out correctly, the reduced angle deviations corresponding to different calibrations will have equal components represented by the first three brackets in expression (8). The difference can only be due to the way the reflecting faces are registered by the autocollimator. Therefore, the difference between the reduced deviations will result in the following:

$$\overline{\Delta\beta_i}^1 - \overline{\Delta\beta_i}^2 = \left[ \Delta a_i^{W_{RS2}} - \frac{1}{N} \cdot \sum_{i=1}^{N} \Delta a_i^{W_{RS2}} \right] - \left[ \Delta a_i^{W_{RS1}} - \frac{1}{N} \cdot \sum_{i=1}^{N} \Delta a_i^{W_{RS1}} \right] \qquad (9)$$

where $a_i^{W_{RS1}}$ and $a_i^{W_{RS2}}$ are the angle deviations of the same face defined by its non-flatness and the apertures that were applied. It is important to note that the resulting difference depends not only on the current face deviation, but also on the change in the averaged deviations of the rest of the faces. This factor has to be accounted for while processing the measurement results.

Ultimately, expression (9) demonstrates that the proposed experiment is in agreement with the goals set in the beginning of the section. After eliminating the unknown internal autocollimator aberrations, the final angular values consist only of members that are defined by the way the non-flat reflecting surfaces are limited by certain apertures. Given that the topography of the polygon faces and the aperture parameters are known, these members may be evaluated by using the suggested model, which gives an opportunity to verify it. In 2009, an experiment that met the presented description was carried out jointly with our colleagues from the "Length and angle graduations" working group (PTB, Germany) at their laboratory. A 24-sided polygon which was previously chosen for international comparison measurements, was selected as a subject of calibration. The angle comparator WMT 905 with an angle resolution of 0.035 arc seconds and a positioning accuracy of 0.03 arc seconds was used as the index table. The optical connection to the polygon faces was provided by the Moeller Wedel Elcomat HR autocollimator. The reflecting face topography was measured using the Moeller Wedel Interferometer V-100 phase-shifting interferometer.

After installing and adjusting the polygon on top of the angle comparator, the calibration was performed on the polygon with fully exposed reflecting faces and with several apertures inserted. The polygon faces were rectangular with dimensions of 20 × 25 mm, and the apertures that were applied were circular with diameters of 20, 18 and 16.5 mm. For each instance, angle measurements were conducted repeatedly during multiple revolutions of the polygon in order to reduce random error impact by averaging. The number of revolutions varied slightly around 40. Direct topography measurement involved saving raw measurement data (point-by-point height profile of the reflecting surface) and was conducted only for fully exposed polygon faces. Further, the statistical parameters (RMS, peak-to-valley, etc.) and Zernike expansions of each topography were evaluated via software processing. The same measurement data for the faces with reduced reflecting area were obtained not through direct measurement, but by the processing of the initially measured full topographies that were algorithmically trimmed by the corresponding virtual masks. The processing of the open reflecting faces was conducted by subtracting the general inclination (the 2nd and 3rd Zernike polynomials). Yet, this parameter was involved in the processing of the masked topographies in order to observe how the least squares approximating plane inclines due to the reflecting face trimming. The measurement protocols of the interferometer for the first polygon face are presented in Appendix A as an example (Figures A1–A4).

The results of this experiment have already been reported previously [26]. However, these were analyzed based on the old assumption that the influence of non-flatness may be accounted for by its statistical parameters. As a part of the current research, in order to verify the proposed model, the old data were analyzed once again. Based on the raw topography data on the faces and the local normals' approximation method, the difference between the reduced angular deviations (9) was calculated for each face of the 24-sided

polygon for the cases of the 20 mm and 16.5 mm apertures. In addition, this difference was calculated based on the polygon calibration data obtained using the autocollimator with the corresponding apertures. Fully exposed faces were not considered in this comparison in order to avoid the influence of possible edge defects of the reflecting faces. The maximum and minimum were selected from the three apertures that were used, since the transition from one to the other provides the most noticeable change in the topography of the faces. The differences obtained are shown in Figure 9.

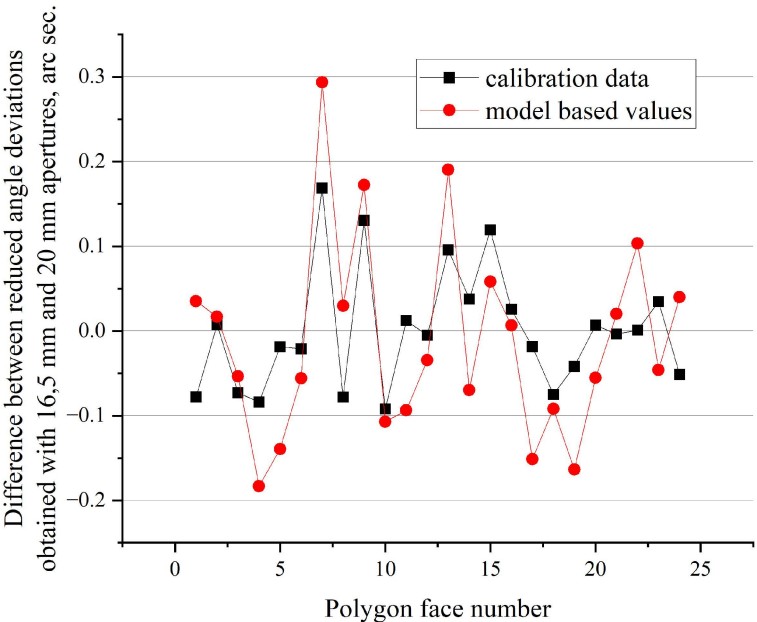

**Figure 9.** The experiment results and model results compared.

To estimate the correlation between these two value sets, the Pearson correlation coefficient was calculated and found as 0.7 with a *p*-value below 0.0002. Although these numbers indicate good correlation, they are still not as good as the authors anticipated them to be. The probable reasons for this are discussed in the next section.

## 5. Discussion

Initially, the model was based on the assumptions that the reticle is a point light source and that the position of its image in the focal plane is determined as an illumination center of mass. Those assumptions justified the application of the local normals' approximation method in determining the effective plane, the angular position of which was measured by the autocollimator. In fact, industrially manufactured contemporary autocollimators (in particular, the Elcomat HR that was used in the experiment) mostly utilize complex multi-slit reticles. The same is true for the evaluation algorithms of the reticle image position, as they are much more sophisticated than simply finding the illumination center of mass. Those complicated reticles and algorithms were developed primarily to increase the autocollimator's sensitivity and to decrease its random error, as well as to solve a range of specific problems, such as the influence of the image distortion due to different measurement conditions (long distance to SUT, small SUT, random aperture stop, etc.). However, despite all the benefits these improvements provide, these also negate the theoretical understanding of the measured object unless the reflecting surface is perfectly flat. There are no other comprehensive methods used to assign the effective plane to the SUT. Mainly for this reason, the suggested model did not provide the desired result. The described problem of losing the theoretical understanding of the measured angle is not critical for the major part of autocollimator applications, in which measurement accuracy does not exceed 0.1 of an arc second. In addition, the problem does not arise if the relative angular deviation of the same surface (or the same area of surface) is being measured. There is no need to determine

what is being tested if the deviation is the only object of interest. However, this problem may be vital for the tasks where the absolute angular position of a reflecting surface has to be registered with high precision (such as optical polygon calibration). Without a theoretical understanding of the measured object, it is impossible to build a model that could take into account the entire range of external factors that affect the autocollimator measurements. This places the accuracy limit on the comparison measurements conducted under slightly different conditions, and the value of this limit cannot even be assessed.

One of the solutions to the stated problem could be creating an autocollimator that would be in agreement with the suggested model. However, it is very unlikely that the manufacturers will revise their autocollimator manufacturing technologies. Besides the fact that it would be a time- and resource-consuming undertaking, there is also a range of problems that need addressing:

- Point light source is a theoretical notion, which in practice may be reached to a certain extent;
- Having a small-size image relatively to a CCD matrix pixel will increase the random error of measurement and decrease sensitivity;
- The vignetting of the reflected beam at the off-axis measurement range intervals will distort the illumination distribution of the reticle image; therefore, the illumination center of mass will not give the adequate image position;
- The uniformity of the exit beam illumination is a serious challenge;
- The assumption that a nearly flat wavefront does not alter while travelling through free space does not apply to long distances and small apertures.

It is worth noting that the measurement conditions from the last entry in the list are in total agreement with the most high-end and challenging angle measurement problem that was mentioned previously, i.e., the deflectometric measurement of the synchrotron optics. In relation to this objective, the problem of increasing the accuracy of autocollimator angle measurement is being solved in the only possible way when the measurement model is unavailable. Multiple metrological facilities are currently devising methods to calibrate the autocollimator at all possible measurement conditions in a reasonable time [27]. Fortunately, the aperture which is placed in front of the reflecting surface for this purpose stays constant and small, and the main altering parameter is the distance to the SUT. Indeed, a single universal solution probably does not exist to overcome the accuracy limit of the autocollimator measurement established by the variance of multiple measurement conditions. Each specific challenge involving the autocollimator will demand such drastic measures as trying to calibrate this device at all possible conditions. However, as for the optical polygon calibration, the authors suggest paying attention to dynamic angle measuring techniques [28] involving an autocollimating null-indicator [29,30]. Without going into detail regarding dynamic angle measurement, the autocollimating null-indicator resembles a regular autocollimator. But instead of measuring angles, it generates the logical pulse at the moment when the reflecting surface is normal to its optical axis. The advantage of this type of device over the conventional autocollimator is that it fits the suggested model almost perfectly. Instead of a reticle, there is a small circular aperture, and this null-indicator processes the analog signal of a photodetector performing the evaluation of the illumination center of mass. In addition, it generates an output signal only when the reflecting surface is normal to the optical axis, which brings the $\varphi$ parameters from (4) to zero and prevents the reflected beam from vignetting. In conclusion, a combination between the dynamic angle measurement and the model suggested by the authors may be promising, and further investigations are to be conducted.

## 6. Conclusions

The autocollimator has had a major part in high precision angle measurements over the past decades, and the need for creating an accurate measurement model has been noticeable for some time. This paper presents the authors' effort to compile an extensive overview on past and present research held in the field of angle metrology, which would substantiate this need and give an insight on the essential measurement parameters. The theoretical aspect of the angular position of the non-flat reflecting surface has been considered. The definition of the surface position, that consists in averaging the local normals, was selected as the closest to the autocollimator operation principle. This definition was used to develop the autocollimator measurement model. The development itself consisted in accounting for the measurement parameters, while averaging the surface local normals based on the wave optics approach. Among the considered parameters were the distance to the reflecting surface, the lateral and axial position of the aperture stop, the size of the autocollimator clear aperture, and the dimensions and shapes of the reflecting surface and the aperture stop. The influence of the internal imperfections in the autocollimator was introduced as an arbitrary wavefront aberration distribution located at its objective lens.

In order to estimate the accuracy level at which the model application would be reasonable, a few simulation cases were presented. Although each specific measurement setup should be assessed separately, the authors suggest that, based on the simulation results, the non-flatness of the reflecting surface, along with other setup properties, have to be accounted for at measurement accuracies higher than 0.2 arc sec. The experimental verification of the model was attempted by analyzing the calibration data on the 24-sided optical polygon. The data were obtained by applying different aperture stops. The authors deemed the verification results as partially satisfactory and presented their considerations as to why, as well as on the model limitations in general.

Considering all the above-mentioned difficulties in taking into account external and internal factors affecting the operation of the autocollimator, the authors believe that its use in metrological routines requiring angular measurement accuracy higher than 0.1 angular second is inexpedient. The possibility of using a high-resolution angular encoder instead of an optical polygon as a transfer standard for measurement accuracies not worse than 0.002 arc second has been shown earlier [31]. In this case, the autocollimator is not required. If the use of an optical polygon is necessary, the authors believe that it is more reasonable either to resort to a dynamic method of angular measurement or to use a Fizeau interferometer instead of an autocollimator.

**Author Contributions:** Conceptualization, supervision and editing, Y.V.F.; theory, simulation, experiments, analysis and writing, R.A.L. All authors have read and agreed to the published version of the manuscript.

**Funding:** The authors are grateful to the Russian Science Foundation for funding within the Grant # 20-19-00412.

**Institutional Review Board Statement:** Not applicable.

**Informed Consent Statement:** Not applicable.

**Data Availability Statement:** Restrictions apply to the availability of these data. Data was obtained from dimension metrology working group at PTB and the availability status is unknown.

**Acknowledgments:** The authors are grateful to R.D. Geckeler, A. Just and the entire dimension metrology working group at PTB, Germany for the opportunity to work at their facility, assistance in experimental data acquisition and fruitful discussions.

**Conflicts of Interest:** The authors declare no conflict of interest.

## Appendix A

<u>Base data</u>

**Protocol from**          05.11.2009    14:55:13
**Measurement from**    23.10.2009    11:35:47
**Examiner:**            Larichev
**Ident-No.**             sp24MWO_face1
**Remark:**              Position (upper left): x - 174 pix. y - 20 pix.
                              Width: 431 pix. Heigth: 523 pix.

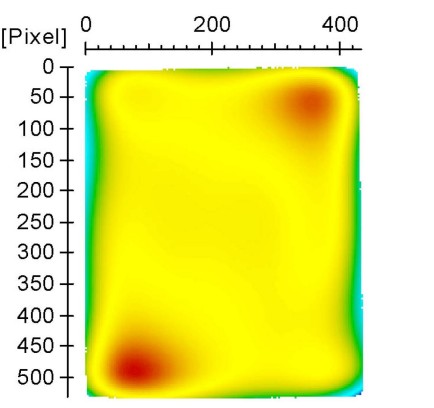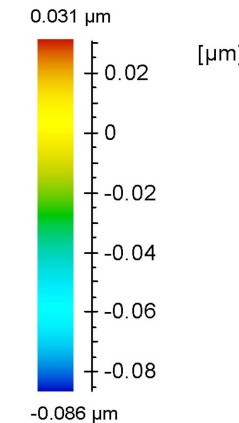

<u>Removed Seidel coefficients</u>
Const Tilt

<u>Measurement results</u>
**Surface form tolerance:** 3 / 0.034   ( 0.413 / 0.114 ) RMSt = 0.047 ; RMSi = 0.046 ; RMSa = 0.044
**P-V:**  0.117 μm
**RMS:**  0.013 μm
**Strehlratio:**  0.984
**Wavelength:**  632.80 nm
**ISO-Wavelength:**  546.07 nm
**Wedge:**  0.500

<u>Zernike coefficients</u>

| | | | |
|---|---|---|---|
| $C_0$=0.000 μm | $C_1$=–0.000 μm | $C_2$=–0.000 μm | $C_3$=–0.005 μm |
| $C_4$=–0.029 μm | $C_5$=0.018 μm | $C_6$=–0.005 μm | $C_7$=–0.001 μm |
| $C_8$=–0.011 μm | $C_9$=0.010 μm | $C_{10}$=–0.002 μm | $C_{11}$=–0.005 μm |
| $C_{12}$=–0.005 μm | $C_{13}$=0.001 μm | $C_{14}$=–0.001 μm | $C_{15}$=–0.001 μm |
| $C_{16}$=–0.088 μm | $C_{17}$=–0.007 μm | $C_{18}$=0.007 μm | $C_{19}$=–0.003 μm |
| $C_{20}$=0.009 μm | $C_{21}$=–0.002 μm | $C_{22}$=–0.001 μm | $C_{23}$=0.002 μm |
| $C_{24}$=–0.001 μm | $C_{25}$=–0.015 μm | $C_{26}$=–0.007 μm | $C_{27}$=–0.060 μm |
| $C_{28}$=0.001 μm | $C_{29}$=0.005 μm | $C_{30}$=–0.004 μm | $C_{31}$=0.017 μm |
| $C_{32}$=0.003 μm | $C_{33}$=–0.002 μm | $C_{34}$=0.001 μm | $C_{35}$=–0.001 μm |

<u>Seidel coefficients</u>
**Focus:** –0.009 μm
**Radius of curvature:**  –81762 m
**Astigmatism:** 0.068 μm   ž=74.24°
**Coma:** 0.015 μm   ž=–165.50°
**Spherical:** –0.067 μm

**Figure A1.** The topography of the fully exposed first face of the optical polygon.

<u>Base data</u>

| | |
|---|---|
| **Protocol from** | 10.11.2009     12:20:29 |
| **Examiner:** | Larichev |
| **Ident-No.** | sp24MWO_face1 |
| **Remark:** | Position (upper left): x - 174 pix. y - 20 pix. |
| | Width: 431 pix. Heigth: 523 pix. |

20 mm aperture in the center

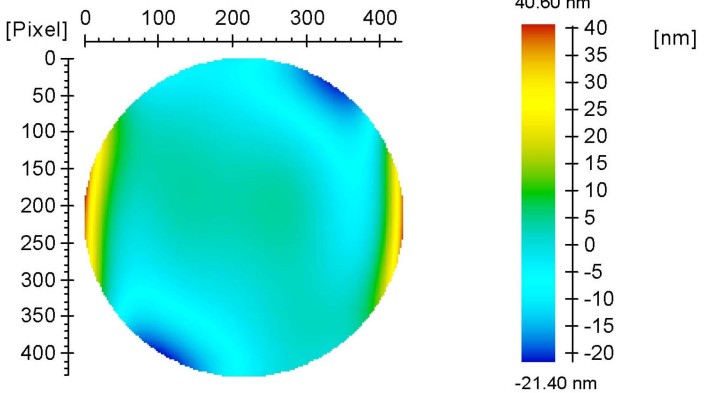

<u>Removed Seidel coefficients</u>

Const

<u>Measurement results</u>

**Surface form tolerance:** 3 / 0.006   ( 0.229 / 0.035 ) RMSt = 0.028 ; RMSi = 0.028 ; RMSa = 0.026
**P-V:**  62.01 nm
**RMS:**  7.53 nm
**Strehlratio:**  0.994
**Wavelength:**  632.80 nm
**ISO-Wavelength:**  546.07 nm
**Wedge:**  0.500

<u>Zernike coefficients</u>

| | | | |
|---|---|---|---|
| $C_0$=0.02 nm | $C_1$=–1.41 nm | $C_2$=–0.28 nm | $C_3$=–0.82 nm |
| $C_4$=11.16 nm | $C_5$=–9.19 nm | $C_6$=0.45 nm | $C_7$=–0.78 nm |
| $C_8$=4.71 nm | $C_9$=–1.76 nm | $C_{10}$=0.31 nm | $C_{11}$=8.24 nm |
| $C_{12}$=–0.77 nm | $C_{13}$=1.27 nm | $C_{14}$=0.77 nm | $C_{15}$=2.12 nm |
| $C_{16}$=7.85 nm | $C_{17}$=1.72 nm | $C_{18}$=0.91 nm | $C_{19}$=–0.21 nm |
| $C_{20}$=3.80 nm | $C_{21}$=0.78 nm | $C_{22}$=–0.50 nm | $C_{23}$=–0.03 nm |
| $C_{24}$=–0.06 nm | $C_{25}$=1.20 nm | $C_{26}$=0.78 nm | $C_{27}$=4.01 nm |
| $C_{28}$=–0.10 nm | $C_{29}$=–0.18 nm | $C_{30}$=0.18 nm | $C_{31}$=–0.49 nm |
| $C_{32}$=–0.08 nm | $C_{33}$=0.05 nm | $C_{34}$=–0.01 nm | $C_{35}$=0.01 nm |

<u>Seidel coefficients</u>

**Tilt:** 1.43 nm    ž=–168.83°
**Focus:** –1.64 nm
**Radius of curvature:**  –187207 m
**Astigmatism:** 28.90 nm    ž=–19.73°
**Coma:** 2.70 nm    ž=–59.87°
**Spherical:** 28.25 nm

**Figure A2.** The topography of the first face of the polygon trimmed by a 20 mm mask.

<u>Base data</u>

| | |
|---|---|
| **Protocol from** | 10.11.2009    11:52:13 |
| **Examiner:** | Larichev |
| **Ident-No.** | sp24MWO_face1 |
| **Remark:** | Position (upper left): x - 174 pix. y - 20 pix. |
| | Width: 431 pix. Heigth: 523 pix. |

18 mm aperture in the center

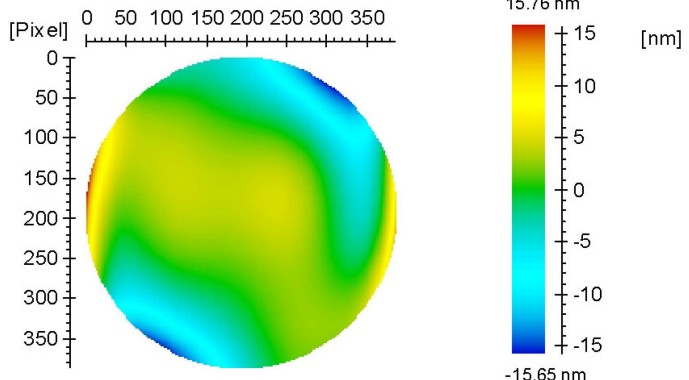

<u>Removed Seidel coefficients</u>
Const

<u>Measurement results</u>
**Surface form tolerance:** 3 / 0.024   ( 0.111 / 0.014 ) RMSt = 0.016 ; RMSi = 0.015 ; RMSa = 0.014
**P-V:**   31.41 nm
**RMS:**   4.48 nm
**Strehlratio:**   0.998
**Wavelength:**   632.80 nm
**ISO-Wavelength:**   546.07 nm
**Wedge:**   0.500

<u>Zernike coefficients</u>

| | | | |
|---|---|---|---|
| $C_0$=−0.01 nm | $C_1$=−1.57 nm | $C_2$=−0.07 nm | $C_3$=−3.29 nm |
| $C_4$=4.34 nm | $C_5$=−7.21 nm | $C_6$=−0.17 nm | $C_7$=−1.00 nm |
| $C_8$=1.73 nm | $C_9$=−1.66 nm | $C_{10}$=0.33 nm | $C_{11}$=3.04 nm |
| $C_{12}$=−1.00 nm | $C_{13}$=1.20 nm | $C_{14}$=0.42 nm | $C_{15}$=1.12 nm |
| $C_{16}$=2.55 nm | $C_{17}$=1.17 nm | $C_{18}$=0.71 nm | $C_{19}$=−0.25 nm |
| $C_{20}$=2.36 nm | $C_{21}$=0.45 nm | $C_{22}$=−0.23 nm | $C_{23}$=−0.07 nm |
| $C_{24}$=−0.06 nm | $C_{25}$=0.74 nm | $C_{26}$=0.43 nm | $C_{27}$=2.11 nm |
| $C_{28}$=−0.04 nm | $C_{29}$=−0.09 nm | $C_{30}$=0.09 nm | $C_{31}$=−0.21 nm |
| $C_{32}$=−0.03 nm | $C_{33}$=0.02 nm | $C_{34}$=−0.00 nm | $C_{35}$=0.01 nm |

<u>Seidel coefficients</u>
**Tilt:** 1.57 nm    ž=−177.53°
**Focus:** −6.58 nm
**Radius of curvature:**   −37606 m
**Astigmatism:** 16.83 nm    ž=−29.48°
**Coma:** 3.03 nm    ž=−99.64°
**Spherical:** 10.39 nm

**Figure A3.** The topography of the first face of the polygon trimmed by an 18 mm mask.

<u>Base data</u>
**Protocol from**    10.11.2009    10:04:15
**Examiner:**    Larichev
**Ident-No.**    sp24MWO_face1
**Remark:**    Position (upper left): x - 174 pix. y - 20 pix.
    Width: 431 pix. Heigth: 523 pix.
16.5 mm aperture in the center

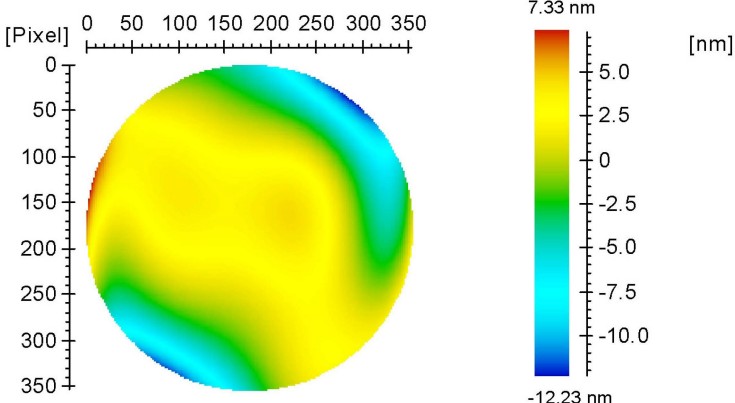

<u>Removed Seidel coefficients</u>
Const

<u>Measurement results</u>
**Surface form tolerance:** 3 / 0.027   ( 0.066 / 0.006 ) RMSt = 0.013 ; RMSi = 0.010 ; RMSa = 0.010
**P-V:**   19.57 nm
**RMS:**   3.56 nm
**Strehlratio:**   0.999
**Wavelength:**   632.80 nm
**ISO-Wavelength:**   546.07 nm
**Wedge:**   0.500

<u>Zernike coefficients</u>
$C_0$=0.01 nm $\quad\quad$ $C_1$=−1.59 nm $\quad\quad$ $C_2$=0.10 nm $\quad\quad$ $C_3$=−3.67 nm
$C_4$=1.97 nm $\quad\quad$ $C_5$=−5.76 nm $\quad\quad$ $C_6$=−0.72 nm $\quad\quad$ $C_7$=−0.98 nm
$C_8$=0.61 nm $\quad\quad$ $C_9$=−1.69 nm $\quad\quad$ $C_{10}$=0.35 nm $\quad\quad$ $C_{11}$=0.82 nm
$C_{12}$=−0.94 nm $\quad$ $C_{13}$=0.83 nm $\quad\quad$ $C_{14}$=0.34 nm $\quad\quad$ $C_{15}$=0.69 nm
$C_{16}$=0.62 nm $\quad\quad$ $C_{17}$=0.85 nm $\quad\quad$ $C_{18}$=0.46 nm $\quad\quad$ $C_{19}$=−0.19 nm
$C_{20}$=1.53 nm $\quad\quad$ $C_{21}$=0.32 nm $\quad\quad$ $C_{22}$=−0.18 nm $\quad$ $C_{23}$=0.02 nm
$C_{24}$=−0.04 nm $\quad$ $C_{25}$=0.47 nm $\quad\quad$ $C_{26}$=0.31 nm $\quad\quad$ $C_{27}$=1.25 nm
$C_{28}$=−0.02 nm $\quad$ $C_{29}$=−0.05 nm $\quad$ $C_{30}$=0.05 nm $\quad\quad$ $C_{31}$=−0.10 nm
$C_{32}$=−0.02 nm $\quad$ $C_{33}$=0.02 nm $\quad\quad$ $C_{34}$=−0.01 nm $\quad$ $C_{35}$=0.01 nm

<u>Seidel coefficients</u>
**Tilt:** 1.59 nm    ž=176.36°
**Focus:** −7.34 nm
**Radius of curvature:**   −28373 m
**Astigmatism:** 12.18 nm    ž=−35.57°
**Coma:** 3.64 nm    ž=−126.12°
**Spherical:** 3.68 nm

**Figure A4.** The topography of the first face of the polygon trimmed by a 16.5 mm mask.

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
