# Peer review of "A Model of Angle Measurement Using an Autocollimator and Optical Polygon"

_photonics, doi:10.3390/photonics10121359_

Round 1
Reviewer 1 Report (Previous Reviewer 1)
Comments and Suggestions for Authors
The revised version is publishable.
Author Response
Dear reviewer,
thank you for your positive opinion.
Reviewer 2 Report (Previous Reviewer 3)
Comments and Suggestions for Authors
Authors are asked to avoid any more complicated sentence structure that a main sentence and a single dependent sentence. A sentence in English does not start with a proposition. A pendant sentence does not proceed a main sentence. All occur several times in the MS, making reading somewhat difficult.
Can you provide some affirmative conclusions on the basis of your work?
I would in the future prefer non-marked/ highlighted manuscript on which I can write my comments using computer software.
Comments on the Quality of English LanguageSee above.
Author Response
Dear reviewer,
thank you for your notes, we tried to improve several sentences in the text and also we added positive conclusion. All new corrections are highlighted by blue color.
This manuscript is a resubmission of an earlier submission. The following is a list of the peer review reports and author responses from that submission.
Round 1
Reviewer 1 Report
Comments and Suggestions for Authors
This paper analyzed thoroughly the error factors and angle measurement principle that affect the measurement of angle using an autocollimator, proposed an autocollimator angle measurement model, and proved that the average of the surface local normal is the optimal definition for angle measurement. In addition, the author proposes a dynamic and static combination method. The proposed model is novel and has positive significance in improving the measurement accuracy of the autocollimator.
Other comments are as follows:
1) In line 152, the formula is incomplete.
2) In lines 203-204, 'Intra pixel quantum...' should have the same format as the previous line.
3) In lines 251-252, it is better to provide Zernike aberration polynomials here.
4) In line 322, The abbreviation 'PSF' should be explained.
Comments on the Quality of English LanguageThe quality of english language is good.
Reviewer 2 Report
Comments and Suggestions for Authors
The paper presents the authors’ effort to compile the historical overview and the proposed studies in angular metrology to give an insight on essential measurement factors affecting on the autocollimator. I would recommend the following items to support clearly your model and its experimental corroboration.
1) Experimental topological data of 24-sided optical polygon and its local normals average effective plane calculation.
2) Wavefront aberration measurement data of autocollimator objective lens which is key component and its sensitivity, if possible.
Reviewer 3 Report
Comments and Suggestions for Authors
The work does not require editing service, just more clear and simple writing with diagrams.